# The Value of White Cell Inflammatory Biomarkers as Potential Predictors for Diabetic Retinopathy in Type 2 Diabetes Mellitus (T2DM)

**DOI:** 10.3390/biomedicines11082106

**Published:** 2023-07-26

**Authors:** Ana Maria Dascalu, Dragos Serban, Denisa Tanasescu, Geta Vancea, Bogdan Mihai Cristea, Daniela Stana, Vanessa Andrada Nicolae, Crenguta Serboiu, Laura Carina Tribus, Corneliu Tudor, Adriana Georgescu, Mihail Silviu Tudosie, Daniel Ovidiu Costea, Dan Georgian Bratu

**Affiliations:** 1Faculty of Medicine, Carol Davila University of Medicine and Pharmacy Bucharest, 020021 Bucharest, Romaniageta.vancea@umfcd.ro (G.V.);; 2Ophthalmology Department, Emergency University Hospital Bucharest, 050098 Bucharest, Romania; 3Department of Nursing and Dentistry, Faculty of General Medicine, ‘Lucian Blaga’ University of Sibiu, 550169 Sibiu, Romania; 4Faculty of Dental Medicine, Carol Davila University of Medicine and Pharmacy Bucharest, 020021 Bucharest, Romania; 5Faculty of Medicine, Ovidius University Constanta, 900470 Constanta, Romania; 6General Surgery Department, Emergency County Hospital Constanta, 900591 Constanta, Romania; 7Faculty of Medicine, University “Lucian Blaga”, 550169 Sibiu, Romania; 8Department of Surgery, Emergency County Hospital Sibiu, 550245 Sibiu, Romania

**Keywords:** diabetic retinopathy, biomarkers, neutrophil-to-lymphocyte ratio (NLR), platelet-to-lymphocyte ratio (PLR), monocyte-to-lymphocyte ratio (MLR), systemic inflammation index (SII)

## Abstract

The pathogenesis of diabetic retinopathy is still challenging, with recent evidence proving the key role of inflammation in the damage of the retinal neurovascular unit. This study aims to investigate the predictive value of the neutrophil-to-lymphocyte ratio (NLR), platelet-to-lymphocyte (PLR), lymphocyte-to-monocyte ratio (LMR), and systemic inflammation index (SII) for diabetic retinopathy (DR) and its severity. We performed a retrospective study on 129 T2DM patients, divided into three groups: without retinopathy (NDR), non-proliferative DR (NPDR), and proliferative DR (PDR). NLR, MLR, and SII were significantly higher in the PDR group when compared to NDR and NPDR (3.2 ± 1.6 vs. 2.4 ± 0.9 and 2.4 ± 1.1; *p* = 0.005; 0.376 ± 0.216 vs. 0.269 ± 0.083 and 0.275 ± 0.111, *p* = 0.001; 754.4 ± 514.4 vs. 551.5 ± 215.1 and 560.3 ± 248.6, *p* = 0.013, respectively). PDR was correlated with serum creatinine (OR: 2.551), NLR (OR: 1.645), MPV (OR: 1.41), and duration of diabetes (OR: 1.301). Logistic regression analysis identified three predictive models with very good discrimination power for PDR (AUC ROC of 0.803, 0.809, and 0.830, respectively): combining duration of diabetes with NLR, MLR, and, respectively, PLR, MPV, and serum creatinine. NLR, MPV, SII, and LMR were associated with PDR and could be useful when integrated into comprehensive risk prediction models.

## 1. Introduction

Diabetes mellitus is a major health problem globally, accounting for 537 million patients worldwide and with an expected ascendant trend reaching 700 million by 2045 [1,2]. For type 2 diabetes mellitus (T2DM), which accounts for approximately 90% of the total, this rising trend can be attributed to aging, a rapid increase in urbanization, and obesogenic environments. Insulin resistance and low-grade systemic inflammation lead to multiple organ damage by microvascular and macrovascular complications. Despite significant achievements in early diagnosis and therapy, diabetic retinopathy is currently the major cause of blindness and visual impairment in working-age adults worldwide [2], causing an increased burden on national healthcare systems worldwide [3]. The prevalence of diabetic retinopathy is estimated to be 27.0% in diabetic patients, which leads to 0.4 million incidences of blindness in the world [4]. The World Health Organization (WHO) Universal Eye Health: A Global Action Plan 2014–2019 outlines the need to achieve a reduction in the prevalence of avoidable visual impairment and blindness, including that related to diabetes, which is currently among the five most common causes of both moderate or severe visual impairment and blindness [5]. DR is listed as a priority eye disease in the 2030 IN SIGHT strategy [6]. However, an effective screening may be challenging due to limited available retina specialists. Finding new biomarkers with potential predictive value may be a valuable tool in the management of diabetic patients by identifying early those at risk for the development of sight-threatening complications.

Chronic hyperglycemia induces both neurodegeneration and apoptosis of the retinal ganglion cells and photoreceptors and causes microvascular retinal damages by multiple pathways: activation of aldose-reductase and polyol pathway, activation of the protein kinase C (PKC) pathway, accumulation of reactive oxygen species (ROS), nitric oxide deficiency, and changes in the blood flow, with increased platelet adhesion and aggregation [7,8]. Nonenzymatic glycation and glycoxidation of proteins lead to the accumulation of advanced glycation end products (AGEs) in the extracellular matrix and subendothelial space, leading to basal membrane thickening, pericyte loss and monocyte migration, and activation of nuclear factor (NF)-κB along with activation of pro-inflammatory pathways [9,10]. Animal studies proved that monocytes use pigment epithelium (RPE) as a gateway for trafficking to the retina [11].

The role of ocular and systemic inflammation in the pathogenesis of diabetic retinopathy has been confirmed by many previous articles [12,13,14]. Impaired glucose and lipid metabolism lead to the accumulation of toxic metabolites and pro-inflammatory changes in the retina, leading to damage to the neurovascular unit [15,16]. Several inflammatory markers such as interleukin (IL)-1, IL6, IL8, transforming growth factor β1, and tumor necrosis factor-α have been linked to end organ damage in diabetes [17,18,19]. However, their practical use is limited due to the high cost and lack of availability in clinical practice.

Blood cell changes contribute to the pathophysiological changes observed in diabetic retinopathy through various mechanisms, including endothelial dysfunction, leukocyte adhesion and infiltration, cytokine and chemokine production, platelet activation, and neovascularization [20]. Increased levels of systemic inflammation were correlated with activated circulant platelets, presenting bigger and inequal volumes, resulting in higher MPV (mean platelet volume) and PDW (platelets distribution width) values [21,22]. Similarly, destruction and fragmentation of the erythrocytes in an inflammatory environment, with endothelial activation, sludge, and microcapillary occlusion, was associated with higher RDW (red cell distribution width) values [23].

White cell inflammatory biomarkers, such as neutrophil-to-lymphocyte ratio (NLR), platelet-to-lymphocyte (PLR), lymphocyte-to-monocyte ratio (LMR), and systemic inflammation index (SII), can be easily calculated based on the complete blood cells count (CBC). In recent studies, they showed good predictive value for several inflammatory, oncologic, and cardiovascular diseases [24,25]. Previous research showed higher values in diabetic patients when compared to normal subjects. Moreover, higher values of NLR and PLR were found to be predictive for diabetic microvascular and macrovascular complications, such as diabetic foot ulcer and the risk of amputation [2], diabetic nephropathy [26] and coronary artery disease (CAD), and end-organ damage in type2 diabetes mellitus (T2DM) [17].

However, there is still conflicting evidence regarding the significance of these biomarkers in clinical evaluation and individual management of diabetic patients. There are relatively wide differences reported for the “cut-off” values that could be predictive of diabetic retinopathy and its severity.

In this paper, we analyzed the correlations between diabetic retinopathy and NLR, PLR, MLR, and SII. Also, we analyzed the RDW and MPV values in T2DM patients with noDR comparatively, with non-proliferative DR, and with proliferative DR. The potential predictive value of these biomarkers was studied both for the onset of any changes of DR in T2DM patients and the presence of proliferative DR (PDR). 

## 2. Materials and Methods

A 1-year retrospective study was performed on patients with type 2 diabetes mellitus (T2DM) admitted to the Ophthalmology Department of Surgery, Emergency University Hospital Bucharest, for cataract surgery, between January 2022 and December 2022. Data were collected from observation charts and electronic patient files.

Age, sex, duration of diabetes, and associated pathologies were documented for each patient. Each patient underwent a complete ophthalmological exam. The diagnosis and staging of diabetic retinopathy were performed according to the Guidelines of the International Council of Ophthalmology for Diabetic Eye Care [27,28]. The patients included in the study group were classified based on the fundoscopic findings into the noDR group (NDR), non-proliferative DR group (NPDR), and proliferative diabetic retinopathy group (PDR).

Blood routine tests and complete blood count with differential were performed after collecting fasting peripheral blood from each patient. Systemic inflammatory indices NLR, PLR, and MLR were calculated as the ratios of the neutrophils, platelets, and monocytes to lymphocytes, respectively. The systemic inflammatory index (SII) was calculated by the formula SII = P × N/L, where P, N, and L are the count for platelets, neutrophils, and lymphocytes, respectively [29]. All counts were determined from the same automated blood sample measurement and expressed as a value in cells/L.

Patients with systemic or ocular inflammatory conditions and oncologic or previously diagnosed hematologic conditions were excluded. From the remaining patients, paired individuals in terms of age and sex were selected to be included in the 3 study groups (NDR, NPDR, and PDR), to limit the effect on the final statistical analysis of these two potential cofounders [29].

### Statistical Analysis

MedCalc^®^Statistical Softwareversion 22.006 (MedCalcSoftwareLtd., Ostend, Belgium; https://www.medcalc.org; accessed on 5 June 2023) was used for statistical analysis. An ANOVA test was used for continuous variables. For the statistically significant results, a post hoc analysis was performed to establish the differences within the three groups by using the Tukey–Kramer test for all pairwise comparisons. A Pearson chi-square test, Fisher’s exact test, and Linear by Linear test were used to evaluate the association between discrete variables.

The specificity and sensitivity of NLR, PLR, MLR, and SII in predicting DR and PDR were analyzed by ROC curves. A minimum value of 0.6 area under the curve (AUC) was considered as a criterion for an acceptable discrimination model [30]. Logistic regression analysis was used to study correlations between several biological parameters and DR. The best regression models were compared by an ROC AUC (area under the curve) score in terms of efficiency of prediction.

## 3. Results

A total of 129 Caucasian patients with T2DM were divided into three study groups according to the presence and the severity of DR: NDR (36 patients), NPDR (49 patients), and PDR (44 patients). The general data of the patients included in the study group are presented in Table 1. 

There were no statistically significant differences in age, sex ratio, associated diseases, lipidic profile, fasting blood glucose (FBG), Hb, and HbA1C between the three study groups. The duration of diabetes was significantly different among the three groups, being well correlated with the severity of DR. The RDW distribution was different among groups. However, the differences were not statistically significant in the post hoc analysis.

Serum creatinine was significantly different among the study groups (*p* = 0.024). The Tukey–Kramer test showed significantly higher values in PDR when compared to the NDR group. The proportion of patients with abnormal serum urea and creatinine values was significantly higher in the PDR group (*p* = 0.035; *p* = 0.013, respectively), with no differences between NDR and NPDR groups. These data suggest a higher prevalence of associated retinal and renal microvascular damages in the PDR group. The patients in the PDR group were also associated with more diabetic micro- and macrovascular complications, when compared to NDR and NPDR groups; however, the differences were not statistically significant.

When analyzing blood cell counts for different types of white cells, only neutrophils were significantly higher in the PDR group when compared to NDR and NPDR groups (*p* = 0.007). While platelet count was not significantly different among the three groups, MPV was higher in the PDR group when compared to NDR and NPDR groups, suggesting an increased platelet activation in the proliferative stage of DR.

Furthermore, white cell inflammatory biomarkers were comparatively analyzed (Table 2). 

PLR values increased along with the severity of DR. However, the differences among groups were not statistically significant. NLR, MLR, and SII indices were significantly higher in the PDR group (*p* = 0.005; *p* = 0.001; *p* = 0.013, respectively) when compared to NDR and NDPR. The post hoc analysis shows no differences between NDR and NPDR groups. 

### 3.1. ROC Curves and Predicting the Value of White Cell Inflammatory Biomarkers

For the systemic inflammatory indices that showed statistically significant different distribution among the study groups after post hoc analysis (MPV, NLR, SII, and LMR), the predictive power for DR and PDR was further assessed by ROC curves.

None of the studied biomarkers met the criteria of a minimum AUC ROC of 0.6 for predicting DR in the study group. When the prediction value for PDR was analyzed, NLR, LMR, and SII showed good discrimination power with high specificity and low sensitivity (Table 3).

### 3.2. Logistic Regression Approach

The binary logistic regression model was performed to analyze the correlation of PDR (1 = true or 0 = false) with each independent variable listed in Table 1 and Table 2. The variables with a *p*-value of <0.05 and the minimal value of the confidence interval for each OR > 1 were selected as possible risk factors for PDR (Table 4), ensuring that each risk factor adds a 95%-significant extra hazard. 

A higher correlation with PDR was encountered for creatinine (OR: 2.551), NLR (OR: 1.645), MPV (OR: 1.41), LMR (OR: 161.19), and duration of diabetes (OR: 1.301). Age, FBG, HbA1C, PLR, RDW, and urea were not significantly correlated with PDR.

Patients with diabetes present a diverse array of comorbidities and metabolic dysregulation features that may combine and accelerate the progression to proliferative diabetic retinopathy. A multivariate logistic regression analysis was further carried out to identify the predictive models with the best discrimination value based on the analyzed variables (Table 5).

We described three different models for predicting PDR based on different systemic inflammatory biomarkers. One model combines NLR and duration of diabetes, while the second combines PLR, MPV, serum creatinine, and duration of DM and the last MLR and duration of diabetes. All models proved a very good discrimination power for PDR, with an AUC ROC of 0.803, 0.830, and 0.809, respectively. The described models show a superior prediction when compared to each of the variables taken separately (*p* < 0.05), based on comparative ROC curve analysis (Figure 1).

Combining SII with different other parameters resulted in a logistic regression model with slightly lower AUC ROC, compared with NLR (0.784 vs. 0.803; *p* = 0.23) and MLR (0.784 vs. 0.830, *p* = 0.17).

## 4. Discussion

Monocytes, neutrophils, and platelets play a significant role in the pathology of diabetic retinopathy. While considered a metabolic disease initially, more and more evidence points out the role of inflammation in retinal damage. A large array of cytokines and chemokines, including monocyte chemotactic protein-1 (MCP1), interleukin-6 (IL-6), interleukin-8 (IL-8), tumor necrosis factor-α (TNF-α), and interferon-γ, were found to be elevated in serum and vitreous of DR patients [31,32]. Activated platelets are larger and display more enzymatic activities. Along with their key role in coagulation and thrombosis, platelets can release a large array of mediators of inflammation, regulating the leukocytes and endothelial cells’ activity [33].

Several studies investigated the clinical value of the white cell fractions and ratios but with conflicting results. A large survey by Wan et al. [34], comparing 2709 patients with no retinopathy with 512 patients with DR, found no differences in neutrophils and platelets number but lower monocytes in early stages of DR. The lymphocyte-to-monocyte ratio (LMR) was considered a good indicator of the circulating immune status of the host in several oncologic diseases [35,36]. LMR was correlated with serum levels of interleukin-6 (IL-6), tumornecrosisfactor, IL-1β, and monocyte chemotactic protein 1, which was noticed in higher levels in vitreous and serum in patients with PDR [37]. Thus, MLR could be a cheaper biomarker to reflect the level of inflammatory changes in patients with DR.

There are only a few studies that analyzed the value of MLR in DR. Yue et al. [32] found a limited value for MLR in the diagnosis of DR and PDR, with NLR and PLR being much more relevant. However, other studies [37,38] proved a significant positive correlation between MLR values and PDR. Huang et al. [37] found a good predictive value for MLR in discriminating between T2DM patients with DR and those with no complication, with an AUC ROC of 0.868, which we could not find in the present study. One explanation may be the increased proportion of PDR cases included in the DR group (64%). We found similar MLR values in NDR groups with the study of Huang et al. [37] (0.27 ± 0.08 vs. 0.269 ± 0.083) but a slightly higher cut-off value for PDR (0.30 vs. 0.36).

The neutrophil-to-lymphocyte ratio (NLR) and platelet-to-lymphocyte ratio (PLR) are promising biomarkers for evaluating the inflammatory status in diabetic retinopathy (DR). However, to maximize their clinical utility, it is essential to establish the appropriate classification systems or cut-off values for NLR and PLR concerning the presence and severity of DR. The NLR values in the normal adult population vary between 0.43 and 3.53, in previous studies, with an average value of 1.56 [39], with slightly higher values in females versus males [40]. PLR is a newly explored biomarker which was correlated with cardiac diseases and all-cause mortality. An ample epidemiological study performed by Wu et al. [40] in a Chinese population found that PLR ranged between 36.63 and 149.13 in healthy males and between 43.36 and 172.68 in women. Moosazadeh et al. [41] found a mean PLR of 110.84 ± 56.53 and 123.12 ± 36.21 in Iranian healthy males and females, respectively. In a prospective cohort study including 8711 adults aged over 45 from the Rotterdam area, Fest and colleagues [42] described the mean value and 95% reference intervals for NLR, PLR, and SII of 1.76 (0.83–3.92), 120 (61–239), and 459 (189–1168), respectively (Fest). Previous studies showed a good prognostic value of PLR in oncologic and cardiovascular diseases [43]. In a large study on27,321individuals, the mean PLR was significantly higher in individuals who died during the follow-up period comparative with those who survived (145.7 vs. 133.0) [43].

Several studies have attempted to establish cut-off values for NLR and PLR to differentiate between patients with and without DR or to stratify patients according to DR severity. However, the proposed cut-off values have varied considerably across studies, reflecting differences in patient populations, study designs, and statistical methods [44,45].

Zeng et al. [31] found significant differences between T2DM with no DR and with DR for NLR and PLR, but no NLR or PLR was associated with the severity of retinopathy. However, in the study group only 24 patients with PDR vs. 124 with NPDR were included. Zeng et al. [31] found, however, that in a multivariate regression analysis, PLR was significantly correlated with the existence of DR, together with diabetic peripheral neuropathy, systolic blood pressure, and duration of diabetes. 

In a study by Ilhan et al. [46], NLR was significantly higher in PDR versus healthy controls (2.67 ± 1.02 vs. 1.85 ± 0.49). An NLR value of 2.11 or more predicted PDR or severe NPDR with a sensitivity of 76% and a specificity of 80%.

Hu et al. analyzed the predictive value of NLR for the response to anti-VEGF therapy in DR patients and found that an NLR  <  2.27 showed a better improvement in letter scores than those with NLR  >  2.27 [47]. A recent study by Wang and colleagues [48] comparing T2DM patients with and without DR found that NLP is an independent risk factor for DR (OR: 1.37; 95%CI: 1.06, 1.78). Similar to our findings, although PLR was not independently associated with DR as a continuous variable, including PLR in a multivariate analysis improved the discrimination power of the statistical model [48].

In a systematic review by Luo et al. [33], MPV mean values ranged between 7.76 and 8.12 in NDR vs. between 8.18 and 10.76 in PDR groups, while the mean NLR had a range of 1.54–2.4 for NDR vs. 1.91–2.58 in PDR patients. The differences may be related to the methodology of research and classification of DR but also patients’ age, BMI, comorbidities, and geographical and ethnic differences. Moreover, the relationship between NLR and DR appears to be non-linear [49], and some studies have suggested that changes in NLR and PLR may be more informative when analyzed as continuous variables rather than categorical variables based on arbitrary cut-off values [2,50,51]. T2DM patients often present with a unique but diverse series of risk factors, comorbidities, and complications, which may accumulate over time due to an inappropriate lifestyle [52]. For instance, a higher NLR or PLR may be associated with an increased risk of DR or more severe DR, but the precise risk may depend on other factors, such as glycemic control, duration of diabetes, and the presence of other microvascular or macrovascular complications [53,54].

While multiple studies proved that SII is an important indicator of systemic inflammation as well as for the risk of acute cardiovascular events, there is little evidence regarding the value range of SII in a normal population. A recent paper of Bai et al. [55] found a mean value of 374 (153, 832) in non-pregnant women.In a large epidemiological study of Luo et al. [56], the mean SII was 334, with a 95% CI of 142–804, with significantly higher values in females and at younger age.SII was investigated as a predictive biomarker for survival in various oncological and cardiovascular diseases. Feng et al. [57] found a cut-off value of 410 × 10^9^ cells/L as a prognostic tool in estimating 5-year survival in patients with squamous cell carcinoma of the esophagus, while Hirara et al. [58] showed that low SII values <661.9 are corelated with overall survival in gastric cancer. A large study on 42,875 adults found that adults with SII levels of >655.56 had higher all-cause mortality and cardiovascular mortality than those with SII levels of <335.36 [59].Guo et al. [60] found a significantly higher SII in patients with T2DM with chronic kidney disease (634.14 ± 13.43 vs. 546.42 ± 10.13). There is also previous evidence that SII might be correlated with the level of retinal damage in T2DM. Eybeli et al. [61], in a recent study from 2022, found that SII and duration of diabetes may predict the incidence of diabetic macular edema (DME) in a group of patients with NPDR. He found a significantly higher value of SII in DME patients vs. the non-DME group (599.7 ± 279.2 and 464. 9 ± 172.2, respectively).

T2DM is associated with chronic inflammatory changes, higher levels of cytokines, and increased oxidative stress, which lead to neurodegeneration and destruction of blood vessels, causing damage to multiple organs [62]. This explains the higher values observed for all three study groups in our study, varying from 551.5 ± 215.1 × 10^9^ cells/L in non-DR patients to 754.4 ± 514.4 × 10^9^ cells/L for the proliferative DR group. This finding may be explained by the multiple micro- and macrovascular diabetes-related comorbidities often encountered in these very vulnerable patients. We also found a high cut-off value for SII in our study, of 763.8, but with a fair predictive value (AUC = 0.623) for PDR. We consider that, while not particularly specific for PDR, higher values of SII are useful in clinical management of DR patient, as an overall mirror of the level of inflammation and vascular damage.

In our study, we found that no statistical differences were noticed between NDR and NPDR groups for any of the white cell inflammatory biomarkers. However, significantly higher values for NLR, MLR, SII, and MPV were found in the PDR group when compared with NDR and NPDR groups. This finding may signify that the level of systemic inflammation is higher in the advanced stage of DR associated with neovascularization. These findings support the idea that diabetic retinopathy is a complex, multifactorial process. Systemic inflammation is one of the key elements that need to be further investigated. However, by combining multiple data points, the power of prediction increases and allows individualized management of each case.

The number of patients enrolled in this study is one of the main limitations of this study. This study did not evaluate the correlation of systemic inflammatory biomarkers with the level of macular edema, which can also be a significant cause of visual impairment in NPDR patients. This research is retrospective, based on the standard protocol of paraclinical exams used for patients admitted for cataract surgery. Data regarding the patients’ medication were not available and were not taken into account in the statistical analysis. Several biological data were not evaluated, such as BMI, systolic and diastolic blood pressure, and the lipidic profile. 

Systemic inflammatory biomarkers offer valuable information regarding the level of inflammation and vascular frailty in diabetic patients, and these two elements should be taken into account in evaluating the risk for developing PDR. The main shortcoming in the context of the current knowledge is the relatively wide range for normal values of these parameters. One future approach may be a dynamic evaluation of the changes in NLR, PLR, MPV, and SII analyzed with the correlation of DR progression. Further studies, including multiple risk factors evaluation, with diverse patient populations could identify better predictive models for clinical use. Such a model may provide a more accurate and personalized assessment of the inflammatory status and DR risk in individual patients with diabetes mellitus [63].

## 5. Conclusions

This paper brings new evidence that supports the role of chronic systemic inflammation in the pathology of diabetic retinopathy. Systemic white cell inflammatory biomarkers did not predict DR in our study group. However, they proved to be of clinical value in assessing PDR, reflecting better the changes associated with the proliferative diabetic retinopathy than each of the white cell count differential taken separately. There are cheap, inexpensive tools that can be valuable in clinical practice. Higher values of NLR, LMR, SII, PLR, and MPV are significantly correlated with PDR in T2DM patients. The best predictive value was obtained for NLR and MLR when combined with the duration of diabetes. Platelet-derived biomarkers (MPV and PLR) may be useful in evaluating the risk of PDR when correlated with other clinical and biological data.

## Figures and Tables

**Figure 1 biomedicines-11-02106-f001:**
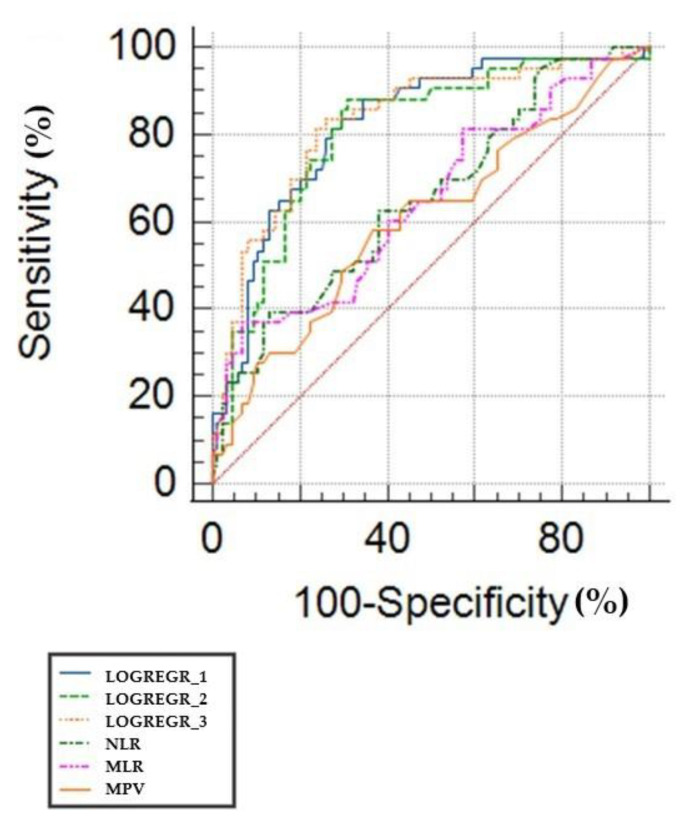
Comparative ROC curves for the three multivariate models described in Table 5, MLR, NLR, and MPV. The red line signifies a specificity and sensitivity of 50%.

**Table 1 biomedicines-11-02106-t001:** General data of the patients included in the study group.

	Total	NDR	NDPR	PDR	*p*-Value
N	129	36	49	44	
Age (mean ± SD)	65.6 ± 8.9	67 ± 6.6	65.2 ± 7.1	63 ± 9.4	0.078 *
Males (n, %)	67 (51%)	16 (44.4%)	23 (46.9%)	26 (59%)	0.393 **
Duration of diabetes (yrs.)	8.9 ± 3.8	5.3 ± 2.4	9.36 ± 3.3	11 ± 3.1	<0.001 *
Associated DM complications:					
Diabetic kidney disease	13 (10%)	3 (8.3%)	2 (4%)	8 (18.1%)	0.072 **
Peripheral arterial disease	11 (8.5%)	2 (5.5%)	3 (6.1%)	6 (13.6%)	0.325 **
Diabetic foot	4 (3.1%)	1 (2.7%)	1 (2%)	2 (4.5%)	0.778 **
Associated diseases (n, %):					
Arterial hypertension	47	20	8	19	0.518 **
Ischemic cardiopathy	4	2	1	1	0.563 **
Glaucoma	5	1	2	2	0.916 **
FBG (mean ± SD)	164.6 ± 60.3	145.5 ± 41.4	173.6 ± 78.6	170.4 ± 46	0.078 *
HbA1C (mean ± SD)	7.6 ± 1.6	7.2 ± 1.1	7.5 ± 1.8	8.2 ± 1.8	0.113 *
Hb (mean ± SD)	13.5 ± 1.6	13.6 ± 1.3	13.5 ± 1.6	13.4 ± 1.2	0.646 *
RDW (mean ± SD)	13.9 ± 1.3	14 ±1.1	13.7 ± 1.4	14.2 ± 1.3	0.031 *
Neutrophils (mean ± SD)	4.9 ± 1.4	4.8 ± 1.4	4.6 ± 1.2	5.4 ± 1.5	0.007 *
Lymphocytes (mean ± SD)	2.1 ± 2.8	2.1 ± 0.7	2.2 ± 0.8	1.9 ± 0.7	0.285
Monocytes (mean ± SD)	0.5 ± 0.1	0.5 ± 0.1	0.5 ± 0.1	0.6 ± 0.1	0.066 *
Platelets (mean ± SD)	237.4 ± 53	224.3 ± 42.6	248.7 ± 46.3	235.3 ± 64.9	0.106 *
MPV mean ± SD)	9 ± 1.1	8.9 ± 1.1	8.7 ± 0.8	9.3 ± 1.2	0.02 *
TG (mean ± SD)	148.3 ± 78.2	136.3 ± 63.4	149 ± 64.2	152.9 ± 92.4	0.076 *
Cholesterol (mean ± SD)	163.19 ± 73.2	158.39 ± 61.7	162.14 ± 53.2	165.9 ± 70.5	0.341 *
Serum urea (mean ± SD)	51.8 ± 28.7	48.4 ± 31.2	49.1 ± 25.8	57.6 ± 29.3	0.264 *
Serum Creatinine	1.1 ± 0.6	0.9 ± 0.3	1 ± 0.5	1.3 ± 0.8	0.024 *
Creatinine > 1.2mg/dL (n, %):	29 (22.4%)	6 (16.6%)	6 (12.2%)	16 (36.3)	0.013 **
Urea > 60 mg/dL (n, %)	30 (23.2%)	7 (19.4%)	8 (16.3%)	15 (34%)	0.035 **

Footnote: * ANOVA; ** Chi-squared test; NDR; nodiabetic retinopathy; NPDR: non proliferative DR; PDR: proliferative DR; FBG: fasting blood glucose; RDW: red cells distribution width; MPV: mean platelet volume.

**Table 2 biomedicines-11-02106-t002:** NLR, PLR, SII, and LMR distribution in the study groups.

	Total	NDR	NDPR	PDR	*p*-Value
NLR (mean ± SD)	2.6 ± 1.3	2.4 ± 0.9	2.4 ± 1.1	3.2 ± 1.6	0.005 *
PLR (mean ± SD) × 10^9^ cells/L	126 ± 54.1	115.4 ± 38.9	122.1 ± 35.4	138.9 ± 76.1	0.127 *
MLR (mean ± SD)	0.308 ± 0.157	0.269 ± 0.083	0.275 ± 0.111	0.376 ± 0.216	0.001 *
SII (mean ± SD) × 10^9^ cells/L	624 ± 365.5	551.5 ± 215.1	560.3 ± 248.6	754.4 ± 514.4	0.013 *

Footnote: * ANOVA; NLR: neutrophil-lymphocyte ratio; PLR: platelet-lymphocyte ratio; LMR: lymphocyte to monocyte ratio; SII: systemic inflammatory index.

**Table 3 biomedicines-11-02106-t003:** Sensitivity and specificity at the “cut-off” value predicting PDR.

	PDR Sensitivity (%)	PDR Specificity (5)	Cut-Off Value	AUC	*p*
NLR	40.0	86.9	>3.18	0.662	0.001
MLR	35.6	92.9	>0.364	0.643	0.006
SII	35.6	85.7	>763.8 (×10^9^ cells/L)	0.627	0.015
MPV	55.6	63.1	>9.24	0.593	0.084
PLR	26.7	91.7	>168.8 (×10^9^ cells/L)	0.536	0.518

**Table 4 biomedicines-11-02106-t004:** Logistic regression model for the dependent variable of PDR.

Risk	Estimated Co-Efficient	Standard Error	Wald	Degrees of Freedom	*p*-Value	OR	Lower	Upper
Duration of diabetes	0.263	0.061	18.55	1	<0.0001	1.301	1.154	1.467
MPV	0.348	0.174	3.984	1	0.045	1.41	1.006	1.994
NLR	0.498	0.165	9.062	1	0.002	1.645	1.189	2.275
MLR × 10	0.508	0.162	9.82	1	0.0017	1.662	1.209	2.284
SII	0.001	0.000	7.23	1	0.007	1.001	1	1.003
creatinine	0.936	0.414	5.11	1	0.02	2.551	1.132	5.746

Footnote: INPUT description: PDR (proliferative diabetic retinopathy) variable is binary (1 = present/0 = absent), and all risks are binary (1 = yes/0 = no)—OR, estimated odd ratio; CI, confidence interval.

**Table 5 biomedicines-11-02106-t005:** Logistic regression models based on multiple variables.

Model No.	Variable	Coefficient	Std.Error	Wald	*p*	OR	95% CI Lower	95% CI Upper
LOGREG_1	NLR	0.46364	0.17296	7.1857	0.0073	1.632	1.156	2.304
	Duration DM	0.28342	0.067991	17.3759	<0.0001	1.314	1.154	1.498
	Constant	−4.59859	0.89211	26.5713	<0.0001			
LOGREGR_2	MPV	0.46284	0.21874	4.4771	0.0344	1.5886	1.0347	2.439
	PLR	0.011012	0.0048070	5.2481	0.0220	1.0111	1.0016	1.0206
	creatinine	0.87851	0.42178	4.3384	0.0373	2.4073	1.0532	5.5025
	Duration of DM	0.30659	0.071052	18.6195	<0.0001	1.3588	1.1821	1.5618
	Constant	−10.14928	2.51771	16.2502	0.0001			
LOGREGR_3	MLR	0.5234	0.1747	8.9763	0.0027	1.6879	1.1984	2.3772
	Duration DM	0.2853	0.0706	16.3062	<0.0001	1.3302	1.1582	1.5278
	Constant	−4.97103	0.9542	27.1372	<0.0001			

## Data Availability

The data presented in this study are available on request from the corresponding author. The data are not publicly available due to privacy.

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
