# Peer review of "The Value of White Cell Inflammatory Biomarkers as Potential Predictors for Diabetic Retinopathy in Type 2 Diabetes Mellitus (T2DM)"

_biomedicines, 2023, doi:10.3390/biomedicines11082106_

Round 1

Reviewer 1 Report

- The PDR for PLR and SII is too high in Table 2 compared with other factors. explain the reason in detail.

- The introduction is not written well. In many cases need to improve the discussion. the following reference can help to improve that part: https://doi.org/10.1016/j.ijbiomac.2022.01.134 

- In table 3 the amount of cut-offs is 763.8 for SII. compared with other values it is showing high value. please consider it.

- Figure 1 needs a unit for sensitivity. Also, the quality of the figure is too low, improving the resolution of the figures in the whole manuscript is necessary.

- I cannot see Figure 2. It is missing.

- if possible, write a prospectives for the future of this research.

Author Response

Dear Reviewer,

Thank you very much for your detailed approach in evaluating our work. We have carefully revised the paper according to your suggestions and comments.

The detailed changes are presented below: 

The PDR for PLR and SII is too high in Table 2 compared with other factors. explain the reason in detail.

R: we have added a paragraph in the Discussion section and commented these results.

- The introduction is not written well. In many cases need to improve the discussion. the following reference can help to improve that part: https://doi.org/10.1016/j.ijbiomac.2022.01.134 

R: We have improved our Introduction and Discussions sections, taking into account the recommended reference.

- In table 3 the amount of cut-offs is 763.8 for SII. compared with other values it is showing high value. please consider it.

R: we have added a paragraph in the Discussion section and commented these results.

- Figure 1 needs a unit for sensitivity. Also, the quality of the figure is too low, improving the resolution of the figures in the whole manuscript is necessary.

R: We have improved the quality of the figure and added the required information

- I cannot see Figure 2. It is missing.

R: We have corrected, it was an error from the initial template used.

- if possible, write a prospectives for the future of this research.

 R: We have added a paragraph with the future perspectives of using systemic inflammatory biomarkers in diabetic retinopathy screening and monitoring.

We do hope that in this revised version you will find our paper suitable to be published.

Reviewer 2 Report

Dear Editor,

I read with interest the article by Dascalu et al. regarding the predictive role of white cell inflammatory biomarkers for diabetic retinopathy in type II DM. The authors enrolled in this study 129 Caucasian patients with T2DM divided into 3 groups according to the presence and the severity of diabetic retinopathy.

The conclusions of this study were that higher values of NLR, LMR, SII, PLR, and MPV are significantly correlated with PDR in T2DM patients. The best predictive value was obtained for NLR and MLR when combined with the duration of diabetes. Platelets-derived biomarkers (MPV and PLR) may be useful in evaluating the risk of PDR when correlated with other clinical and biological data.

The manuscript is well written and of interest. However, the number of patients enrolled in this study is one of the main limitations of this study. It would be very interesting if the authors could specify the medicinal treatment of the patients, and introduce this in the statistical analysis. If not, then this aspect must be specified among the limitations of the study.

Moreover, the authors should present more comorbidities of the patients such as: peripheral arterial disease and chronic kidney disease (given that table 1 shows that in the PDR group we have a higher incidence of patients with elevated values of creatinine and urea ).

Author Response

Dear reviewer,

Thank you very much for your appreciation of our work! We have revised carefully our paper according to your recommendations.

Data regarding the patients’ medication were incomplete in the electronic patient records since they had a very short hospital stay of only 1-2 days for cataract surgery. We included this in the limitations of the study as recommended.

Moreover, the authors should present more comorbidities of the patients such as peripheral arterial disease and chronic kidney disease (given that table 1 shows that in the PDR group, we have a higher incidence of patients with elevated values of creatinine and urea ).

Thank you for your suggestion! We added these data in table 1 and we have commented that in our study: The patients in the PDR group also associated more diabetic micro and macrovascular complications, when compared to NDR and NPDR groups, however, the difference was not statistically significant.

We hope that in this revised version you will find our paper suitable for publication.